# Mixture of Latent Experts Using Tensor Products

**Zhan Su**                                                                *zhan.su@di.ku.dk*
*University of Copenhagen, Denmark*

**Fengran Mo**                                                        *fengran.mo@umontreal.ca*
*University of Montreal, Quebec, Canada*

**Prayag Tiwari**                                                     *prayag.tiwari@ieee.org*
*School of Information Technology, Halmstad University, Sweden*

**Benyou Wang**                                                   *wangbenyou@cuhk.edu.cn*
*The Chinese University of Hong Kong, Shenzhen, China*

**Qiuchi Li**                                                            *qiuchi.li@di.ku.dk*
*University of Copenhagen, Denmark*

**Jian-Yun Nie**                                                        *nie@iro.umontreal.ca*
*University of Montreal, Quebec, Canada*

**Jakob Grue Simonsen**                                           *simonsen@di.ku.dk*
*University of Copenhagen, Denmark*

**Reviewed on OpenReview:** *https://openreview.net/forum?id=SgxeJW4DGk*

## Abstract

In multi-task learning, the conventional approach involves training a model on multiple tasks simultaneously. However, the training signals from different tasks can interfere with one another, potentially leading to *negative transfer*. To mitigate this, we propose a novel *latent-expert* approach (`TensorPoly`), that balances parameter efficiency with nuanced routing methods. For *experts*, we reparameterize Low-Rank Adaptation (`LoRA`) by employing an entangled tensor through the use of tensor product operations and name the resulting approach `TLoRA`. For *routing function*, we tailor two innovative routing functions according to the granularity: `TensorPoly-I` which directs to each rank within the entangled tensor while `TensorPoly-II` offers a finer-grained routing approach targeting each order of the entangled tensor. The experimental results from the multi-task T0-benchmark demonstrate that: 1) all latent-expert approaches surpass the corresponding dense approaches, highlighting the potential of modular language models to mitigate negative inference in multi-task learning and deliver superior outcomes. 2) `TensorPoly-I` achieves higher parameter efficiency in adaptation and outperforms other modular LMs, which shows the potential of our approach in multi-task transfer learning [1].

## 1 Introduction

Recently, the de facto paradigm for natural language understanding (NLU) tasks has centered on leveraging large language models (LLMs) (He et al., 2021) that are pre-trained on a vast corpus of unlabelled data and subsequently fine-tuned for specific tasks (Qiu et al., 2020; Ye et al., 2021). While this approach has significantly advanced the field, it often requires substantial computational resources and may not efficiently

---

[1]The code is released: https://github.com/microsoft/mttl

transfer knowledge across diverse tasks. In addition, fine-tuning tasks independently would lead to *negative transfer*, where the lack of shared information across tasks makes it difficult to achieve compositional generalization (Ponti et al., 2023).

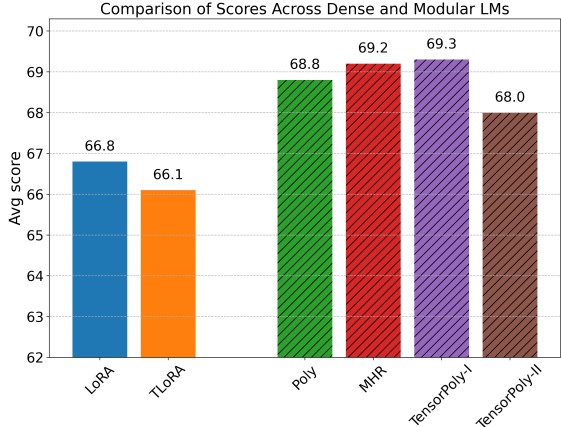 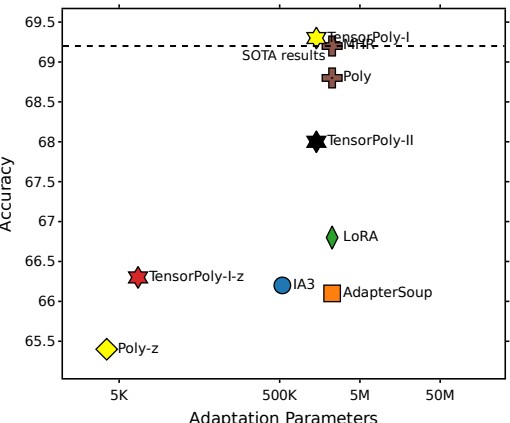

Figure 1: *Left*: Comparison between the dense models (LoRA, TLoRA) and latent-expert approaches (Poly, MHR, TensorPoly-I, TensorPoly-II). Poly/MHR use LoRA as the modules, `TensorPoly-I` and `TensorPoly-II` use TLoRA as the modules. *Right*: Adaptation parameters across different approaches in the fine-tuning process.

To address the aforementioned issues, there have been, roughly, two general lines of research. The *first line* aims to mitigate the computation and memory issue using lightweight alternatives known as parameter-efficient fine-tuning (PEFT), which updates only a small number of extra parameters while keeping most pre-trained parameters frozen (Houlsby et al., 2019b; Li & Liang, 2021; Hu et al., 2021). However, these solutions need to train an adapter for each task, which does not take into account the fact that test tasks may require solving different combinations of sub-problems compared to training tasks (Vu et al., 2020), thus failing to achieve compositional generalization (Rosenbaum et al., 2019; Ponti, 2021).

The *second line* is to facilitate information sharing across multiple tasks using multi-task learning (MTL) approaches (Caruana, 1997; Zhang & Yang, 2021; Liu et al., 2019), which simultaneously train the model on several tasks, allowing it to learn shared representations for all tasks involved. However, MTL necessitates access to all training tasks during the training phase, meaning that incorporating new tasks requires retraining the model from scratch. This requirement significantly increases the computational burden and limits the flexibility of the model to adapt to new tasks efficiently.

A promising approach to address the above issues is the adoption of modular language models (Pfeiffer et al., 2023; Ponti et al., 2023; Caccia et al., 2023), where modules are typically implemented as PEFTs for different tasks (Hu et al., 2021; Houlsby et al., 2019a; Bach et al., 2022). Information flow is conditionally routed to a subset of these modules, which are then aggregated for the given task. This design facilitates the positive transfer and systematic generalization (Pfeiffer et al., 2023). Recently, `Poly` and `MHR` were designed to handle diverse tasks by leveraging different combinations of latent experts (Ponti et al., 2023; Caccia et al., 2023). Given $|\mathcal{T}|$ tasks, there are only $|\mathcal{K}| < |\mathcal{T}|$ experts trained. We treat these $|\mathcal{K}|$ experts as latent experts, and each task-specific adapter can be obtained by a linear combination of latent experts. During both the multi-task pre-trained and fine-tuning, `Poly` implements adapters with LoRAs and concurrently optimizes the LoRA inventory and a routing function. `MHR` partitions the LoRAs into multiple heads and uses a finer-trained routing among these heads. However, LoRA adapters are still limited in parameter efficiency, especially with expert libraries involving a huge number of adapters. In addition, we notice that previous approaches only use linear combinations of experts, which implicitly assume that the given task has a linear relationship with the expert modules, whereas the relations could be much more complicated in practice. To this end, the question of developing a modular language model that balances parameter efficiency with

complex routing methods is critical for advancing the scalability and functionality of multi-task transfer learning.

To answer this question, we devise a new variant `Poly` model: `TensorPoly`, which mixes experts using tensor product (Smolensky, 1990), an operation that maps two vectors in spaces $\mathcal{V}$ and $\mathcal{W}$ to a vector in the tensor product space $\mathcal{V} \otimes \mathcal{W}$ of the two vector spaces, associated with a bilinear map $\mathcal{V} \times \mathcal{W} \to \mathcal{V} \otimes \mathcal{W}$ (§3.3). This process enables the capture of higher-order interactions and structural relationships between input spaces (Kye, 2023). Tensor product has been successful as a strategy for compressing word embeddings (Panahi et al., 2019; Gan et al., 2022). To achieve a higher parameter-efficient adapter, we have reparameterized LoRA (Hu et al., 2021) adapters by employing an "entangled" tensor structure, where the high hidden and intermediate dimensions are decomposed into smaller dimensions in tensor product form. Consequently, the training matrix $M \in \mathbb{R}^{d \times r}$ in LoRA is reparameterized into a finer-grained tensor $\mathcal{L} \in \mathbb{R}^{N \times r \times \lceil \sqrt[N]{d} \rceil \times R)}$, named TLoRA (4.1). This reparameterization allows for a more nuanced manipulation of the model's parameters, facilitating a more efficient adapter process for more complex tasks. The entangled tensor configuration introduces two critical hyper-parameters: the tensor rank (R) and the tensor order (N). Leveraging these parameters, we develop two distinct routing functions designed to select modules on varying levels of granularity. As depicted in Figure 3, `TensorPoly-I` employs a routing mechanism that assigns distribution scores to different tensor ranks, facilitating the selection of modules based on their rank granularity. Further advancing this concept, we propose a more refined routing function, `TensorPoly-II`, which targets even finer-grained tensors as activated modules. Each module is associated with a specific order of the entangled tensor. Once modules are selected via the routing function, they are aggregated through a tensor product operation, enabling a sophisticated and dynamic assembly of modular skills.

To evaluate the effectiveness and parameter efficiency of our approach, we apply our methods against a series of competitive baselines on T0 (Sanh et al., 2021), a widely used benchmark in multi-task transfer learning covering a high variety of language understanding tasks. Our experiments reveal several key insights: *i)* Modular language models such as `Poly`, `MHR` and `TensorPoly` frameworks consistently outperform traditional PEFT approaches LoRA and TLoRA (Figure 1), underscoring the effectiveness of modular LLMs in facilitating positive transfer across multi-task environments. *ii)* `TensorPoly-I` demonstrates competitive results against `Poly` and `MHR`, while simultaneously achieving higher parameter efficiency in adaptation. This efficiency gain highlights the benefits of our tensorized module approach in achieving high performance with lower parameter overhead. *iii)* A comparative analysis between `TensorPoly-I` and `TensorPoly-II` indicates that the latter's finer-grained routing mechanism does not contribute to improved performance. This outcome suggests that while granularity in module selection is valuable (Caccia et al., 2023), there is a complexity threshold beyond which additional granularity may not yield further benefits.

In summary, our contributions are as follows:

- We propose TLoRA that achieves competitive results while only using $\mathcal{O}(N \times r \times \lceil \sqrt[N]{d} \rceil \times R)$ training parameters compared to LoRA $\mathcal{O}(d \times r)$, highlighting our approach's high parameter efficiency.

- We propose a novel modular LM `TensorPoly`, which balances parameter efficiency with tensor product routing. The evaluation results on T0 benchmark demonstrate that `TensorPoly-I` can surpass other strong modular LMs, underscoring the critical role of tensor product routing in scenarios involving multi-task transfer learning.

- By only fine-tuning the routing function. `TensorPoly-I` can surpass the `TLoRA` with only 8.6k training parameters, achieving extremely parameter-efficient fine-tuning.

## 2 Related Work

**Parameter-efficient Fine-tuning**

Parameter-efficient fine-tuning (PEFT) methods facilitate efficient adaptation of LLMs without updating all the training parameters, thereby reducing the memory and computation cost (Zhang et al., 2023b). One kind of PEFT approach focuses on adding *modules* to LLMs, and only these small *modules* will be trained while

the backbone model is kept frozen and shared across tasks. For example, Adapter Tuning inserts small neural modules (adapters) between the layers of the basic model (Houlsby et al., 2019a), whereas Prefix Tuning and Prompt Tuning add tunable vectors to the input or hidden layer of the base model (Li & Liang, 2021; Lester et al., 2021). Another kind of research model is the incremental update of the pre-trained weights in a parameter-efficient way, without modifying the model architecture. Bitfit fixes all training parameters and only fine-tunes the additive bias term (Zaken et al., 2021). Diff Pruning learns a task-specific "diff vector" that extends the original pre-trained parameters. As the number of tasks increases, Diff Pruning only requires storing a small diff vector for each task (Guo et al., 2020). The seminal paper Hu et al. (2021) proposes a method named `LoRA` that parameterizes incremental weights $\Delta$ as a low-rank matrix by the product of the down projector matrix and up projector matrix. `LoRA` achieves comparable or even superior performance to full fine-tuning (Hu et al., 2021). Zhang et al. (2023a) demonstrates that weight matrices in the top layers are more important than those in the bottom layers. They propose Adaptive Low Rank Adaptation (AdaLoRA), a new method that dynamically allocates the parameter budget among weight matrices during LoRA-like fine-tuning (Zhang et al., 2023a). AdaLoRA adjusts the rank of incremental matrices for different layers. Liu et al. (2024) introduces a new approach DoRA to investigate the inherent differences between full fine-tuning and LoRA. Baziotis et al. (2022) finds hyper-adapters are more parameter efficient than regular adapters, reaching the same performance with up to 12 times less parameters.

Compared to the works above, our approach `TLoRA` takes a novel perspective by reparameterzing LoRA using tensor products and utilizing finer-grained tensors as modules.

## Multi-task Learning

The key to MTL is information sharing across tasks. For this purpose, AdapterSoup Chronopoulou et al. (2023) trains each adapter for each domain, and performs weight-space averaging of adapters trained on different domains. Huang et al. (2023) introduce LoRAhub to aggregate the LoRA modules trained on diverse tasks. They first train a group of LoRA modules that are specialized in each task, then randomly select a subset of modules, and finally learn a set of weights to combine these LoRA models using gradient-free optimization. AdapterFusion (Pfeiffer et al., 2020) proposes a two-stage algorithm that leverages knowledge from multiple tasks. Similarly to LoRAhub, a group of task-specific adapters learn to encapsulate the task-specific information, and in the second stage, a fusion layer combines the trained adapters. Ponti et al. (2023) introduces a variable-size module routing mechanism, `Poly`, based on the assumption that each task correlates with a specific subset of latent skills drawn from a comprehensive inventory of modules. Building upon `Poly`, (Caccia et al., 2023) introduces a finer-grained multi-head routing function `MHR` where the experimental findings underscore the significance of the routing function during the pre-training phase. Mixture-of-expert (MoE) methods such as TaskMoE Kudugunta et al. (2021) learn a routing function that allocates modules to tasks end-to-end.

We propose `TensorPoly-I` and `TensorPoly-II`, two novel routing mechanisms based on TLoRA. `TensorPoly-I` is a variant of Poly using TLoRA as the modules. In this setting, each rank of the entangled tensor corresponds to a separate expert. `TensorPoly-II` is a finer-grained routing function targeting each order of the entangled tensor.

## Expert Merging

Once the experts are activated in the forward pass, we need to aggregate their outputs. There is an increasing focus on aggregating adapters from different domains through expert merging. The simplest operation of merging is averaging the weights of different experts, where the weight of each expert is set according to the routing probability generated by the router (McMahan et al., 2017; Choshen et al., 2022; Matena & Raffel, 2022; Chronopoulou et al., 2023; Huang et al., 2023; Muqeeth et al., 2023; Ostapenko et al., 2023; 2024). `Poly` Ponti et al. (2023) uses the latent experts and integrates the experts by averaging the weights. `MHR` Caccia et al. (2023) partitioned the LoRA experts into different heads, which are eventually concatenated to obtain a merged expert.

We devise two tensor product routing functions `TensorPoly-I` and `TensorPoly-II`. Once each expert is activated, a routing function will aggregate the expert weights as an entangled tensor.

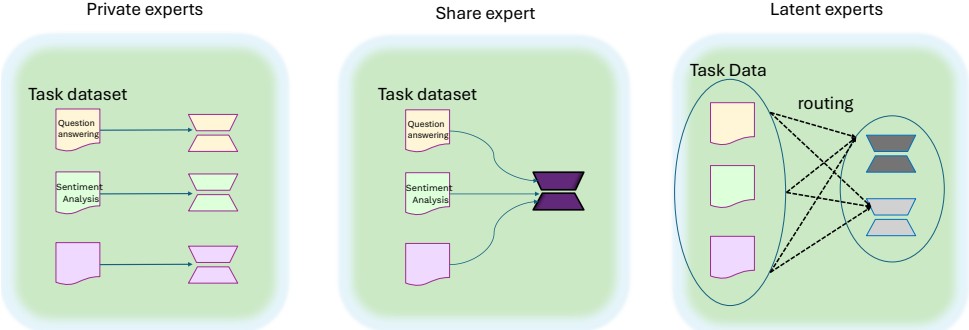

Figure 2: Compare with three training paradigms in multi-task transfer learning. *Left* is the private training, for each task, we train the corresponding expert individually. *Middle* is the shared version, for all the tasks, we train an expert continually, and as a result, we only get one expert. *Right* is the latent experts model, for all the tasks, we train a subset of "latent" experts, so each corresponding expert can be seen as a linear combination of these latent experts.

## 3 Background

In multiple transfer learning tasks, we define a set of tasks as $\mathcal{T} = \{\mathcal{T}_1, ..., \mathcal{T}_{|\mathcal{T}|}\}$. This set is divided into two subsets train $\mathcal{T}_{train}$ and test $\mathcal{T}_{test}$. The goal of multi-task transfer learning is to apply the knowledge from the training tasks $\mathcal{T}_{train}$ to the test tasks within $\mathcal{T}_{test}$. This process involves two main phases. Building on top of a foundation model, the first phase consists of multi-task pre-training using the dataset in tasks $\mathcal{T}_{train}$. The second consists of a few-shot adaptation, where the learned adapters are fine-tuned independently on each test task in $\mathcal{T}_{test}$. We follow the procedure from (Raffel et al., 2020) and formulate each task as a text-to-text problem.

### 3.1 Module: LoRA

Lora is a recently proposed adapter architecture that achieves a competitive balance between performance and parameter efficiency (Hu et al., 2021; Mahabadi et al., 2021). For each linear transformation corresponding to the query ($q$), key ($k$), value ($v$), and output ($o$) of the self-attention layers, LoRA modifies the base model parameters as follows:

$$h = W_0 x + s \cdot A(B)^\top x \tag{LoRA}$$

where $W_0$ are the (frozen) weights of the pre-trained model (e.g. T5 (Raffel et al., 2020)). $A, B \in \mathbb{R}^{d \times r}$ are low-rank learnable parameters and $s \geq 1$ is a tunable scalar hyperparameter. We schematize LoRA in Figure 3.

### 3.2 Polytropon (`Poly`): Mixture of Latent Experts with Linear Combination

`Poly`/ MHR addresses the multi-task problem by softly sharing *latent experts* across tasks. Each `Poly` layer contains 1) an inventory of latent experts $\mathcal{M} = \{\phi_1, \ldots, \phi_m\}$ with $|\mathcal{M}| \ll |\mathcal{T}|$; 2) a routing function $r(\cdot)$ that chooses which subset of the experts to combine for each task. Each latent expert corresponds to a LoRA adapter, where $\phi_i$ are its associated parameters $A^{(i)}, B^{(i)} \in \mathbb{R}^{d \times r}$. $r(\cdot)$ is implemented as a task–module routing matrix $Z \in \mathbb{R}^{|\mathcal{T}| \times |\mathcal{M}|}$. $z_\tau = Z_{\tau,:} \in \mathbb{R}^{|\mathcal{M}|}$ is a routing vector of task $\mathcal{T}_\tau$, with cell $Z_{\tau,j}$ being the probability logits of using module $\phi_j$ for task $\mathcal{T}_\tau$ in the current layer. Differently from mixture-of-experts (Fedus et al., 2022), which perform token-level top-$k$ routing, $Z$ converges to a binary matrix, defining a soft partition over modules. This is achieved by using a Gumbel-sigmoid distribution (Jang et al., 2017) during training, with $\hat{Z}_{\tau,j} \sim \text{Gumbel}(Z_{\tau,j})$. At each forward pass, `Poly` can be defined as:

$$A^\tau = \sum_i \alpha_i A^{(i)}; \ B^\tau = \sum_i \alpha_i B^{(i)} \tag{Poly}$$

where $\alpha_i = \frac{\hat{Z}_{\tau,i}}{\sum_j \hat{Z}_{\tau,j}}$ , and $A^{(i)}, B^{(i)}, A^\tau, B^\tau \in \mathbb{R}^{d \times r}$. We normalize the mixing coefficients $\hat{Z}_{\tau,i}$ for each task to ensure that the number of active modules does not affect the norm of $A^\tau, B^\tau$. Overall, this approach enables different subsets of *modules* to be activated for the current layer and combined in a task-specific way. Following TLoRA, the output of the `Poly` layer is added to the output of the original layer of the frozen backbone: $h = W_0 x + s A^\tau (B^Q)^\top x$.

### 3.3 Tensor, Tensor Product, Entangled Tensor

**Tensor.** The tensor $\mathcal{A}$ is a multi-dimensional array of elements (called components) of $\mathbb{R}$, each being denoted by its integer coordinates in the array; e.g., for a two-dimensional array, the component at position $i, j \in \mathbb{N}$ is denoted $A_{i,j}$. The *order* of a tensor is how many indices it has (e.g., a vector $v$ is a first-order tensor, a matrix $M$ is a second-order tensor, etc.).

**Tensor Product.** The tensor product $\mathcal{V} \otimes \mathcal{W}$ of two vector spaces $\mathcal{V}$ and $\mathcal{W}$ is a vector space to which is associated a bilinear map $\mathcal{V} \times \mathcal{W} \to \mathcal{V} \otimes \mathcal{V}$ that maps a pair of vectors $(v, w), v \in \mathcal{V}, w \in \mathcal{W}$ to a vector in $\mathcal{V} \otimes \mathcal{W}$, denoted as $v \otimes w$. We can create tensor product spaces by more than one application of a tensor product, $\mathcal{H} = \mathcal{U} \otimes \mathcal{V} \otimes \mathcal{W}$, with arbitrary bracketing since the tensor product is associative. The tensor product space of $N$ vector spaces in such form is said to have a tensor *order* of $N$.

$$\bigotimes_{j=1}^{N} \mathcal{H}_j = \mathcal{H}_1 \otimes \mathcal{H}_2 \otimes ... \otimes \mathcal{H}_N \tag{1}$$

**Entangled Tensor.** The vectors in $N$-order tensor product space $\otimes_{j=1}^{N} \mathcal{H}_j$ are in the form $v = \otimes_{j=1}^{N} v_j, v_j \in \mathcal{H}_j$, and referred to as *simple tensors*. Vectors represented as the sum of multiple simple tensors are called *entangled tensors*:

$$\sum_{k=1}^{R} \bigotimes_{j=1}^{N} v_{jk} = \sum_{k=1}^{R} v_{1k} \otimes v_{2k} \otimes ... \otimes v_{Nk} \tag{2}$$

where tensor rank $R$ is the smallest number of simple tensors that sum up to $v$. For example, $\frac{v_{11} \otimes v_{21} + v_{12} \otimes v_{22}}{\sqrt{2}}$ is a tensor of rank 2.

### 3.4 Tensorized Vector using Entangled Tensor

Any training parameter $v \in \mathbb{R}^d$ can be expressed as an entangled tensor of rank $R$ and order $N$ by: $v = \sum_{k=1}^{R} \bigotimes_{j=1}^{N} v_{jk}$. Here, $v_{jk} \in \mathbb{R}^Q$, yielding a resultant vector $v$ of dimension $p = q^N$. If there exists a value of $N$ such that $d = p$, the storage requirements are efficiently managed, consuming only $RNq = O(Rq \log p/q)$ parameters. More generally, we take the smallest possible value $q$ satisfying $q^N > d$, i.e. $q = \lceil \sqrt[N]{d} \rceil$, and cut off the excess part of the generated vector to produce $v \in \mathbb{R}^d$.

For example, we would like to compress a 512-dim vector ($d = 512$) with a tensor of order $N = 3$ and rank $R = 2$. Then we can set the dimension of each small vectors as $q = \sqrt[N]{d} = 8$. In this way, the vector can be represented with a total number of $RNq = 48$ parameters, leading to a significant parameter reduction.

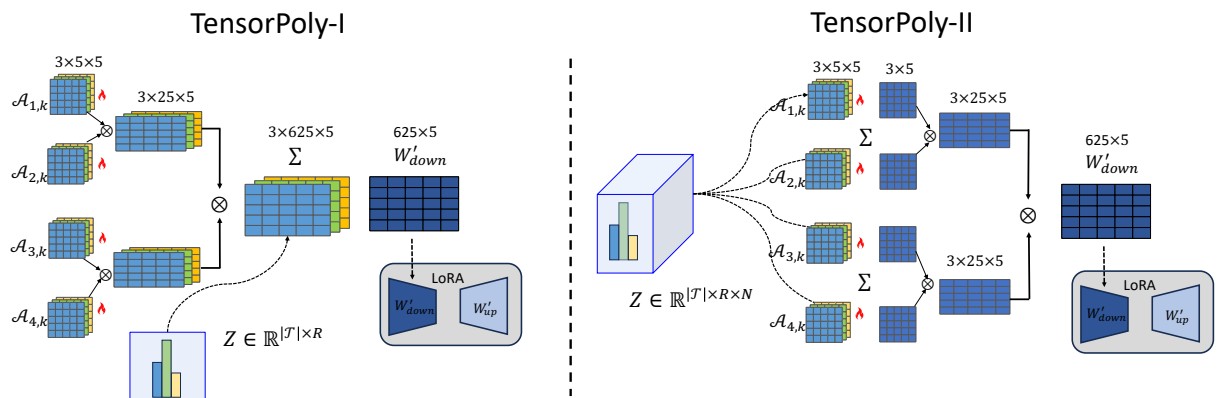

Figure 3: `TensorPoly-I` and `TensorPoly-II`. We illustrate how to reparameterize the LoRA matrix $\mathbb{R}^{625 \times 5}$ with 4 tensor $\mathcal{A} \in \mathbb{R}^{3 \times 5 \times 5}$. In this case, the tensor rank $R = 3$, tensor order $N = 4$. For `TensorPoly-I`, the routing function $\mathbf{Z}$ is designed to select which rank of the entangled tensor is activated for a given task. Conversely, `TensorPoly-II` introduces a more granular control by selecting tensor rank and tensor order.

## 4 Methods: TensorPoly

We propose two variant `TensorPoly` models: `TensorPoly-I` and `TensorPoly-II` based on the TLoRA module (§ 4.1). `TensorPoly-I` employs a routing mechanism that assigns distribution scores to tensor ranks (§ 4.2) while `TensorPoly-II` assigns distribution scores to the finer-grained tensor order.

### 4.1 TLoRA

To achieve a higher parameter efficiency, we reparameterize the LoRA using the tensor product, which is widely used in compressing the word embedding (Panahi et al., 2019; Gan et al., 2022). Essentially, the low-rank matrices $A$ and $B$ are further reparameterized to entangled tensors of rank $R$ and order $N$, following (§3.3). For a given input $x$, `TLoRA` modifies the projection output $h$ as:

$$
\begin{aligned}
h &= W_0 x + s \cdot A(B)^\top x \\
&= W_0 x + s \cdot \left( \sum_{k=1}^{R} \bigotimes_{i=1}^{N} \mathcal{A}_{i,k} \right) \left( \sum_{k=1}^{R} \bigotimes_{i=1}^{N} \mathcal{B}_{i,k} \right)^T x
\end{aligned}
\tag{TLoRA}
$$

where $R$ is the tensor rank, $N$ is the order of entangled tensor respectively. $\mathcal{A}, \mathcal{B} \in \mathbb{R}^{N \times r \times \lceil \sqrt[N]{d} \rceil \times R)}$ are fourth-order tensors, which refer to the training parameters of the modules. $\mathcal{A}_{i,k}, \mathcal{B}_{i,k} \in \mathbb{R}^{r \times \lceil \sqrt[N]{d} \rceil)}$ are indices of fourth-order tensors, referring to specific rank and order in an entangled tensor, as depicted in Fig 3.

### 4.2 TensorPoly: Mixture of Latent Experts using Tensor Products

*Latent-expert* approaches have been proven effective in the few-shot multi-task transfer learning (Ponti et al., 2023; Caccia et al., 2023). `Poly` merges the latent experts by averaging the weight. `MHR` partitioned the LoRA into several heads and use a *piecewise* linear aggregation (i.e., linear within disjoint intervals). Instead of using a linear combination, we propose `TensorPoly` models that incorporate TLoRA as the core module. Specifically, we propose a `TensorPoly-I` and `TensorPoly-II` according to the routing granularity. In each layer, `TensorPoly-I` use a routing matrix $Z \in \mathbb{R}^{|\mathcal{T}| \times R}$ to determine which rank within the entangled tensor can be activated for a given task. In this case, each rank $\mathcal{A}_k$ corresponds to an "expert" in the forward

pass. Upon activation of the selected rank, the model computes a linear combination of the selected rank, weighted by a factor $\alpha$:

$$\mathcal{A}_k = \bigotimes_{i=1}^{N} \mathcal{A}_{i,k} = \mathcal{A}_{1,k} \otimes \mathcal{A}_{2,k} \otimes ... \otimes \mathcal{A}_{N,k}$$

$$A^{\tau} = \sum_{k=1}^{R} \alpha_k \mathcal{A}_k = \underbrace{\alpha_1 \mathcal{A}_1 + ... + \alpha_R \mathcal{A}_R}_{\textbf{merge the rank } R} \qquad \text{(TensorPoly-I)}$$

The same for $B^{\tau} = \sum_{k=1}^{R} \alpha_k \mathcal{B}_k$, where $\alpha_k = \frac{\hat{z}_{\tau,k}}{\sum_j \hat{z}_{\tau,j}}$. The expert $\mathcal{A}_k \in \mathbb{R}^{N \times r \times \lceil \sqrt[N]{d} \rceil}$ is a third-order tensor.

Contrast to the `TensorPoly-I` with a routing matrix, the routing for `TensorPoly-II` is conceptualized as a third-order tensor $\mathbf{Z} \in \mathbb{R}^{|\mathcal{T}| \times N \times R}$, which offers a finer-grained level in directing the model's focus across different ranks and orders of the tensor space. This sophisticated routing mechanism facilitates the selection of finer-grained tensor elements, which are then aggregated through a tensor product operation.

$$A_i^{\tau} = \sum_{k=1}^{R} \alpha_{i,k} \mathcal{A}_{i,k} = \underbrace{\alpha_{i,1} \mathcal{A}_{i,1} + ... + \alpha_{i,R} \mathcal{A}_{i,R}}_{\textbf{merge the rank } R}$$

$$A^{\tau} = \bigotimes_{i=1}^{N} \mathcal{A}_i^{\tau} = \underbrace{\mathcal{A}_1^{\tau} \otimes \mathcal{A}_2^{\tau} \otimes ... \otimes \mathcal{A}_N^{\tau}}_{\textbf{merge the order } N} \qquad \text{(TensorPoly-II)}$$

Each unit of $Z$ is $\alpha_{i,k} = \frac{\hat{z}_{\tau,i,k}}{\sum_j \hat{z}_{\tau,j,k}}$, which will be routed to the specific order and rank in the entangled tensor.

In this case, an "expert" $\mathcal{A}_{i,k} \in \mathbb{R}^{r \times \lceil \sqrt[N]{d} \rceil}$ corresponds to each order in the entangled tensor.

## 5 Experiments

To evaluate the effectiveness of our approaches, we perform experiments on multi-task transfer learning datasets T0 benchmark (Sanh et al., 2021), which is widely used in few-shot generalization approaches. In addition, a diverse array of tasks in T0 benchmark can help us test the generalization ability across different tasks. We conduct a comparative analysis between routing approaches (`Poly`, `MHR`, `TensorPoly-I`, `TensorPoly-II`) and their corresponding "single-expert" (without routing function) version (LoRA,TLoRA), as detailed in §5.2. We also investigate how parameter efficiency and effectiveness are influenced by hyperparameters rank $R$ and order $N$ in an entangled tensor (§5.3.1).

### 5.1 Datasets and Evaluation

**Datasets** To evaluate the generalization capabilities of our models, we adopt the same benchmarking strategy as (Liu et al., 2022), utilizing a subset of tasks designated as held-out from the multitask training. This benchmark encompasses a diverse array of tasks, including sentence completion (COPA (Roemmele et al., 2011), H-SWAG(Zellers et al., 2019) and Story Cloze (Sharma et al., 2018) datasets), natural language inference (ANLI (Nie et al., 2019), CB (De Marneffe et al., 2019) and RTE (Dagan et al., 2005)), coreference resolution (WSC (Levesque et al., 2012), Winogrande (Sakaguchi et al., 2021)), and word sense disambiguation (WIC (Pilehvar & Camacho-Collados, 2018)). For each task, our evaluation strategy involves constructing sets of five few-shot training examples, which are generated by sampling subsets from each dataset using different seeds. We then report the median performance. It should be noted that the prompt examples from each dataset using the prompt templates from P3 (Bach et al., 2022).

| Model | Natural Language Inference | | | | | Sentence Completion | | | Co-reference | | WSD | ACC |
|---|---|---|---|---|---|---|---|---|---|---|---|---|
| | RTE | CB | ANLI1 | ANLI2 | ANLI3 | COPA | H-SWAG | Story | WSC | Wino | WiC | |
| Baselines (w/o pre-train) | | | | | | | | | | | | |
| FullFT | 79.8 | 87.5 | 46.6 | 41.3 | 40.0 | 81.0 | 46.4 | 93.8 | 65.4 | 56.5 | 57.7 | 63.3 |
| BitFit (with LN) | 72.2 | 57.1 | 36.5 | 35.3 | 36.6 | 75.0 | 29.5 | 88.6 | 61.5 | 56.6 | 51.7 | 54.6 |
| LayerNorm | 71.8 | 57.1 | 36.5 | 35.1 | 36.3 | 76.0 | 29.6 | 88.7 | 63.5 | 49.4 | 52.2 | 54.2 |
| Adapter | 76.2 | 87.5 | 45.1 | 40.4 | 35.3 | 84.0 | 41.9 | 91.7 | 65.4 | 54.7 | 55.5 | 61.6 |
| Compacter | 75.8 | 82.1 | 40.8 | 37.4 | 35.8 | 84.0 | 46.4 | 93.5 | 64.4 | 55.5 | 55.2 | 61.0 |
| Compacter++ | 76.9 | 82.1 | 41.7 | 38.3 | 36.9 | 86.0 | 46.3 | 93.5 | 65.4 | 55.1 | 54.1 | 61.5 |
| Prompt(10) | 52.7 | 66.1 | 34.2 | 33.5 | 33.5 | 67.0 | 29.9 | 84.2 | 54.8 | 51.9 | 51.6 | 50.9 |
| Prompt(100) | 48.0 | 53.6 | 33.4 | 33.8 | 33.3 | 60.0 | 26.8 | 74.0 | 60.6 | 51.1 | 50.0 | 47.7 |
| Prefix tuning | 68.6 | 84.0 | 43.3 | 37.5 | 36.5 | 71.0 | 42.1 | 90.2 | 56.7 | 52.0 | 54.2 | 57.8 |
| FishMask (0.2%) | 76.9 | 83.9 | 43.7 | 39.7 | 37.2 | 82.0 | 44.1 | 94.2 | 63.5 | 54.5 | 52.5 | 61.1 |
| FishMask (0.02%) | 75.5 | 76.8 | 39.9 | 38.1 | 36.2 | 84.0 | 38.2 | 93.6 | 61.5 | 53.9 | 53.5 | 59.0 |
| SAID | 69.0 | 80.4 | 40.4 | 35.4 | 35.5 | 77.0 | 36.7 | 89.3 | 61.5 | 52.7 | 55.0 | 57.5 |
| LoRA | 78.3 | 85.7 | 45.1 | 41.0 | 39.5 | 88.0 | 47.1 | 93.6 | 60.6 | 56.8 | 55.2 | 62.8 |
| $(IA)^3$ | 78.0 | 87.5 | 48.6 | 40.8 | 40.8 | 87.0 | 49.4 | 94.7 | 68.3 | 59.8 | 56.0 | 64.6 |
| w/ pre-train | | | | | | | | | | | | |
| LoRA | 81.9 | 89.3 | 41.2 | 40.3 | 41.3 | 93.7 | 59.8 | 96.2 | 66.0 | 67.9 | 56.8 | 66.8 |
| $(IA)^3$ | 82.2 | 89.9 | 45.8 | 41.6 | 41.2 | 91.7 | 53.4 | 94.2 | 70.8 | 63.3 | 53.9 | 66.2 |
| TLoRA | 80.7 | 90.5 | 39.9 | 40.9 | 41.2 | 93.0 | 54.4 | 95.3 | 66.3 | 67.4 | 57.3 | 66.1 |
| Poly | 84.7 | 89.3 | 46.0 | 42.8 | 42.7 | 93.0 | 63.3 | 96.6 | 68.9 | 70.1 | 59.9 | 68.8 |
| MHR | 85.2 | 90.5 | 44.7 | 42.3 | 42.8 | 94.7 | 63.3 | 96.7 | 70.5 | 70.6 | 59.8 | 69.2 |
| TensorPoly-I | 85.2 | 91.7 | 45.0 | 42.5 | 42.5 | 96.7 | 63.1 | 96.6 | 68.6 | 69.8 | 60.6 | **69.3** |
| TensorPoly-II | 84.7 | 90.5 | 44.4 | 41.0 | 42.0 | 94.3 | 58.7 | 95.7 | 67.6 | 68.7 | 59.9 | 68.0 |

Table 1: Results on the T0 few-shot benchmark. All the results in our implementation are the median score of 3 random seeds [0, 1024, 42]. For all the baseline scores, we quote the results from Liu et al. (2022). The value in **bold** is the best score.

**Evaluation** For the evaluation of our models, we employ the rank classification methodology as outlined by the Liu et al. (2022) study. This approach involves ranking the model's log-probabilities for all possible label strings associated with each task. The model's prediction is deemed correct if the label string with the highest log-probability ranking corresponds to the correct answer. This method allows for a nuanced assessment of the model's predictive accuracy by examining its ability to prioritize the correct label over others based on their calculated log-probabilities, offering a precise measure of its understanding and processing of the task at hand.

## 5.2 Baselines

| Method | Pre-Training | Fine-Tuning |
|---|---|---|
| FullFT | $d^2$ | $d^2$ |
| LoRA | $2dr$ | $2dr$ |
| TLoRA | $2Nr\lceil \sqrt[N]{d}\rceil R$ | $2Nr\lceil \sqrt[N]{d}\rceil R$ |
| Poly | $2dr\|S\| + \|\mathcal{T}\|\|S\|$ | $2dr\|S\| + \|S\|$ |
| TensorPoly-I | $2Nr\lceil \sqrt[N]{d}\rceil R + \|\mathcal{T}\|R$ | $2Nr\lceil \sqrt[N]{d}\rceil R + R$ |
| TensorPoly-II | $2Nr\lceil \sqrt[N]{d}\rceil R + \|\mathcal{T}\|RN$ | $2Nr\lceil \sqrt[N]{d}\rceil R + RN$ |

Table 2: Number of parameters (per layer) used for each method. $d$ is the input and output dimension of the training parameters. We assume they are identical. $r$ is the rank in the LoRA, where $r \ll d$. $N$ and $R$ are the order and rank of entangled tensors respectively. $S$ is the number of modules in Poly.

In our comparative analysis, we initially set the benchmark by evaluating the performance of the TLoRA model against the traditional full fine-tuning approach, referred to as FullFT. To facilitate a fair comparison

with baseline methodologies, we have chosen the T0-3B model, consistent with the approach described in the IA3 paper by Liu et al. (2022). In the `FullFT` scenario, we do not freeze any parameters of the pre-trained model, nor do we insert any adapters, allowing for a comprehensive update of the model's parameters during fine-tuning. Subsequently, we contrast our method against a suite of established parameter-efficient fine-tuning (PEFT) baselines to study the efficiency and effectiveness of each in terms of training parameter utilization. These baselines include: **Adapter**, as introduced by Houlsby et al. (2019b), which involves inserting trainable layers while keeping the pre-trained model's parameters fixed; **BitFit** by Zaken et al. (2021), which only fine-tunes the bias terms within the model; **LoRA** proposed by Hu et al. (2021), adjusting the low-rank adaptations of the weight matrices; **Compacter** and **Compacter++** by Karimi Mahabadi et al. (2021), which extend the adapter methodology with compact and efficient training strategies; **Prompt tuning** (Lester et al., 2021) and Prefix tuning Li & Liang (2021) add some tunable vectors to the input or hidden layer of the base model; **FishMask** by Sung et al. (2021), identifying and training a subset of parameters; **Intrinsic SAID** as described by Aghajanyan et al. (2020), focusing on intrinsic sparse activations; and **IA3** (Liu et al., 2022), emphasizing adaptability and efficiency.

We perform a comparative analysis between routing approaches `TensorPoly`, `Poly/MHR` and dense models without routing, specifically TLoRA and LoRA. This comparison aims to evaluate the impact of routing techniques on model performance and efficiency. By contrasting these models, we seek to evaluate how the dynamic allocation of tasks to specific experts in `TensorPoly` and `Poly/MHR` compares to the dense models.

## 5.3    Results

Tab. 1 presents the mean downstream accuracy for 11 held-out tasks in the T0 benchmark. We reported most of the results from (Liu et al., 2022). When evaluating the performance of various PEFT approaches against single-expert performances, it is observed that many PEFT strategies achieve similar outcomes while utilizing a significantly smaller subset of training parameters compared to the FullFT method. Furthermore, the LoRA method within our training framework further illustrates the efficiency of these methods. Specifically, TLoRA achieves a competitive score of 66.1, closely trailing the original LoRA's score of 66.8, while requiring only fewer training parameters used by LoRA. This demonstrates that TLoRA not only matches the effectiveness of LoRA in terms of performance but also achieves a higher parameter efficiency.

In our analysis, we initially contrast the modular model against the dense model, followed by a comparison of routing-based approaches with a single adapter strategy. Within this context, we utilize both LoRA and TLoRA as baselines for these routing techniques. According to the results presented in Tab. 1, the `Poly` model demonstrates superior performance over LoRA by a margin of 2.0 points. Moreover, TensorPoly exhibits an improvement of 3.2 points over the base TLoRA model, underscoring the superiority of routing in enhancing multi-task generalization within multi-task transfer learning.

When evaluating the various modular models, `TensorPoly-I` stands out by not only on par with the recent state-of-the-art achievements but also by outperforming `TensorPoly-II`, despite the latter employing a more granular routing function. This finding is particularly noteworthy, as it suggests that the increased specificity of the routing function in `TensorPoly-II` does not necessarily translate to superior performance. We will discuss this in Section 5.4.

### 5.3.1    Rank and Order Analysis

For `TensorPoly`, rank $R$ and order $N$ correlate with the number of experts. Tab. 2 illustrates how the tensor rank $R$ and the tensor order $N$ affect the training and adaptation parameters. As illustrated in Figure 4 (*left*), when the tensor rank is set to 1, it corresponds to the original TLoRA configuration. An increase in tensor rank necessitates a larger number of training parameters. Consequently, there is a notable enhancement in performance. In addition, we investigate if the rank will give more capacity to the model. As depicted in Figure 4 (*right*), as the increase of the rank, we can get a lower validation loss score during the multi-task pre-training. This progression underscores the direct correlation between the tensor rank and the capability of the model.

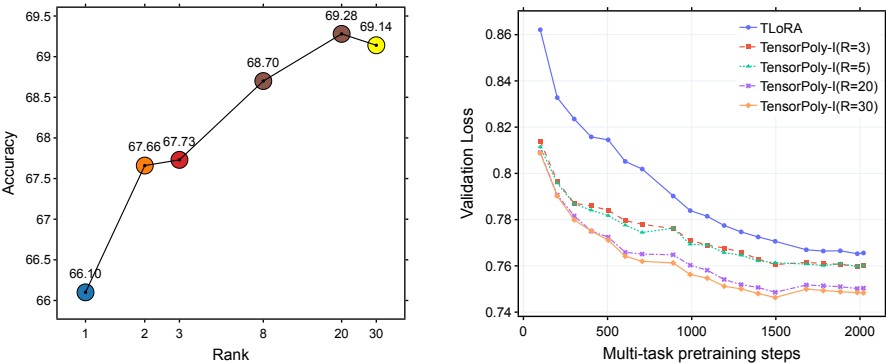

Figure 4: Rank analysis in the `TensorPoly-I`, *Left* denotes the average accuracy over 11 held-out tasks according to different rank. *Right* is the validation loss in the multi-task pre-training process.

The tensor order $N$, correlates with the granularity of training parameters (experts) (§4.2); a tensor of order 4 yields a finer-grained module compared to one of order 2. Our research examines the impact of varying the tensor orders on performance outcomes. For simplicity, we constrain our analysis to tensors of order 2 and 4. Our results demonstrate that a tensor of order 2 outperforms one of order 4 by a margin of 1.7 for the `TensorPoly-I`. For `TensorPoly-II`, an order-2 tensor exceeds the performance of an order-4 tensor by 4.8, suggesting that a balance must be struck between the granularity of experts and its efficacy. Although higher-order tensors may conserve training parameters, this comes at the cost of diminished performance.

| Model | R | N | Multi-Task Params | Adaptation Params | ACC |
|---|---|---|---|---|---|
| LoRA | - | - | 2.2M | 2.2M | 66.8 |
| TLoRA | 1 | 2 | 1.4M | 1.4M | 66.1 |
| Poly | - | - | 17M | 2.2M | 68.8 |
| TensorPoly-I | 8 | 2 | 12.2M | 1.4M | 68.7 |
| TensorPoly-I | 20 | 2 | 27.8M | 1.4M | 69.3 |
| TensorPoly-I | 8 | 4 | 4.3M | 1.4M | 66.9 |
| TensorPoly-II | 8 | 2 | 13.3M | 1.4M | 67.9 |
| TensorPoly-II | 20 | 2 | 33.2M | 1.4M | 68.0 |
| TensorPoly-II | 8 | 4 | 7.6M | 1.4M | 63.1 |

Table 3: Rank $R$ and Order $N$ analysis. The `Adaptation parameter` is the number of parameters required to learn a new downstream task. `Multi-Task params` is the number of additional parameters that must be conserved after multi-task pre-training to enable transfer to a downstream task.

### 5.3.2 Routing Analysis

**Routing Only** In the paper Caccia et al. (2023), the `MHR` study demonstrates that fine-tuning solely the routing function can yield competitive outcomes. This insight provides a valuable perspective for our investigation into various routing strategies within the `TensorPoly` framework. In line with this approach, we focus exclusively on fine-tuning the routing function during the few-shot adaptation process, indicated by the notation $z$. This methodological choice allows us to isolate the impact of the optimization of the routing function on the overall performance of the `TensorPoly` model, thereby offering a clearer understanding of how dynamic routing contributes to the adaptability and efficiency of the model in few-shot learning scenarios.

As detailed in Tab. 4, an initial observation reveals that the routing parameters necessitate a small number of training parameters, achieving extreme parameter efficiency. `TensorPoly-I-`$z$ achieves a accuracy score of 66.3 while only using 8.6k adaptation parameters. `TensorPoly-II-`$z$ achieves 65.4 with 17.3k parameters. `MHR-`$z$ achieve the 68.3 with 220K adaptation parameters. The results indicate that `TensorPoly-I-`$z$ can

| Model | modules | routing | Adaptation Params | ACC |
|---|---|---|---|---|
| LoRA | ✓ | - | 2.2M | 66.8 |
| TLoRA | ✓ | - | 1.4M | 66.1 |
| Modular LMs | | | | |
| Poly-$\mu$ | ✓ | ✗ | 2.2M | 68.8 |
| MHR-$\mu$ | ✓ | ✗ | 2.2M | 68.6 |
| TensorPoly-I-$\mu$ | ✓ | ✗ | 1.4M | 69.0 |
| TensorPoly-II-$\mu$ | ✓ | ✗ | 1.4M | 68.2 |
| Poly-$z$ | ✗ | ✓ | 3.5k | 65.4 |
| MHR-$z$ | ✗ | ✓ | 220K | 68.3 |
| TensorPoly-I-$z$ | ✗ | ✓ | 8.6k | 66.3 |
| TensorPoly-II-$z$ | ✗ | ✓ | 17.3k | 65.4 |
| Poly | ✓ | ✓ | 2.2M | 68.8 |
| MHR | ✓ | ✓ | 2.2M | 69.2 |
| TensorPoly-I | ✓ | ✓ | 1.4M | **69.3** |
| TensorPoly-II | ✓ | ✓ | 1.4M | 68.0 |

Table 4: We compare several fine-tuning approaches. $-\mu$ represents we only fine-tune the modules. $-z$ means we only fine-tune the routing functions. We set the order $N = 2$ for this comparison.

achieve competitive results as LoRA with higher parameter efficiency. Notably, the average accuracy of TensorPoly-I-$z$, and TensorPoly-II-$z$ lag behind their counterparts where both the modules and the routing function undergo fine-tuning. This disparity highlights the critical role that modules play in the fine-tuning process for TensorPoly.

**Modules Only** To verify whether the routing is important in few-shot fine-tuning, we discard the routing function and average the pre-trained modules in the fine-tuning process, indicated by the notation $-\mu$. The result in Tab. 4 shows that there is only a slight decrease for MHR-$\mu$ and TensorPoly-I-$\mu$ compared to their counterparts where both the modules and the routing function are fine-tuned. This result is consistent with the conclusion in Caccia et al. (2023).

## 5.4 Discussion

### 5.4.1 Investigate more about the latent expert approach.

Since we add more training parameters (we use more experts), it is necessary to investigate if the improvement is caused by adding more training parameters. To investigate this, In Caccia et al. (2023), they assign a binary module allocation to each data point in a minibatch, disregarding task information. At test time, the learned modules are averaged into one single module. This approach is named random-$\mu$, the random-$\mu$ performs similar to single LoRA, which proved that the "routing" function is important for latent-expert approach in multi-task transfer learning.

### 5.4.2 Why TensorPoly-II underperform the TensorPoly-I?

Finer-grained routing has been shown to be effective in multi-head routing function (Caccia et al., 2023). However, our experimental results underscore that the finer-grained tensor product routing did not contribute to improved final performance in our models. This discrepancy prompts a future line of inquiry: we plan to explore whether there exist specific benchmarks or conditions under which the tensor-product interaction could demonstrate its purported benefits. Identifying such scenarios will be crucial to harnessing the potential advantages of tensor product routing in modular language models.

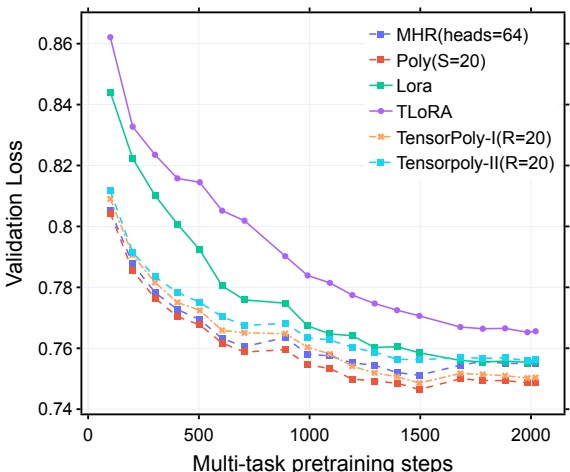

Figure 5: Validation loss compare across different models.

In the current methodology, each tensor-based module is not specialized towards any particular domain. In the future, we intend to explore a more tailored training strategy. This will involve dedicating each tensor to a specific domain and subsequently aggregating these domain-specialized tensors using tensor product operations. Our objective is to assess whether this domain-specific aggregation approach can yield superior generalization capabilities.

# 6 Conclusion

We introduce a novel modular language model named `TensorPoly`. This model incorporates tensorized modules, specifically TLoRA, to significantly reduce the number of training parameters required by the traditional LoRA approach. We employ two distinct strategies for aggregating activated modules: `TensorPoly-I` directs to each rank and a more finely routing, named `TensorPoly-II` targets each order of the tensor. Our evaluation across various multi-task learning scenarios reveals that modular language models, such as `TensorPoly`, surpass the performance of single-adapter models. This underscores the importance of sharing task information through a routing function in multi-task learning contexts. Notably, `TensorPoly-I` achieves state-of-the-art results, highlighting the effectiveness of the `TensorPoly` framework. However, `TensorPoly-II` does not outperform `TensorPoly-I` in our experimental settings, suggesting areas for further investigation in future research.

**Broader Impact Statement**

The research reported in this paper proposes a novel type of language model, and is primarily a theoretical contribution accompanied by experiments to show the practical usefulness on the model, especially in mitigation of negative inference in multi-task learning. Thus, the impact is primarily in generic modelling and empirical performance in situations where existing approaches (e.g., LoRAs) are used, but does not per se create new use scenarios, or raise new concerns, ethical or otherwise, beyond those already present for LoRAs.

**Acknowledgments**

We acknowledge the support of Alessandro Sordoni for optimizing the code and discussion.

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

# A Appendix

## A.1 Dataset

For each task in T0 benchmark(Sanh et al., 2021), there are a few examples to help with the task adaptation. The details are described in Tab 5.

| Dataset | Val Num | Few Shot |
|---|---|---|
| COPA | 100 | 32 |
| WIC | 638 | 32 |
| RTE | 277 | 32 |
| CB | 56 | 32 |
| Winograd | 1267 | 50 |
| RTE1 | 1000 | 32 |
| RTE2 | 1000 | 32 |
| RTE3 | 1200 | 32 |
| WSC | 104 | 32 |
| H-swag | 10042 | 20 |

Table 5: Dataset statistics of the T0 benchmark

## A.2 FLOP Analysis

TLoRA aims to reduce the number of training parameters by parameterizing the original LoRA architecture, but it does not alter the computational process of a LoRA. Thus, TLoRA might conceivably incur additional computational overhead compared to a standard LoRA. To understand this, we analyze the floating-point operations (FLOPs) involved in TLoRA.

Consider the tensor product of a vector $a$ of size $n \times 1$ with a vector $b$ of size $1 \times n$, resulting in a matrix $C$ of size $n \times n$. The product operation involves $n^2$ multiplications due to the $n$ elements in each of the vectors $a$ and $b$. As described in Table 2 and shown in Figure 3, the additional computational effort in TLoRA stems from the tensor product operations, dictated by the order of the tensor. Specifically, for parameterizing with a dimension $d$, rank $r$, and tensor rank $R$, the number of extra computations required is approximately $d \times r \times R$.

To test the time consumption according to the Flop analysis, we show the training time in Tab 6. The results indicate that although the tensor product can reduce the training parameters, we need more training time according to the original LoRA adapters.

| Model | Trainable parameters | Training time | ACC |
|---|---|---|---|
| LoRA | 2.2M | 1d 14h 53m 27s | 66.8 |
| TLoRA | 1.4M | 2d 3h 13m 45s | 66.1 |
| Poly | 17M | 1d 16h 16m 43s | 68.8 |
| MHR | 17M | 1d 17h 5m 20s | 69.2 |
| TensorPoly-I (rank=8) | 12.2M | 2d 3h 13m 8s | 68.7 |
| TensorPoly-II (rank=8) | 13.3M | 2d 3h 48m 19s | 67.9 |
| TensorPoly-I(rank=20) | 27.8M | 2d 6h 6m 3s | 69.3 |
| TensorPoly-II(rank=20) | 33.2M | 2d 2h 55m 59s | 68.0 |

Table 6: Training time analysis across different models.

| Model | Finetuning parameter size | ACC |
|---|:---:|:---:|
| FullFT | 3B | 63.3 |
| BitFiT(with LN) | 1.3M | 54.6 |
| LayerNorm | 250K | 54.2 |
| Adapter | 12.9M | 61.6 |
| Compacter | 807K | 61.0 |
| Compacter++ | 540K | 61.5 |
| Prompt (10) | 41K | 50.9 |
| Prompt (100) | 409K | 47.7 |
| Prefix Tuning | 576K | 57.8 |
| FishMask(0.2%) | 6M | 61.1 |
| FishMask(0.02%) | 600K | 59.0 |
| SAID | 500K | 57.5 |
| LoRA | 9.1M | 62.8 |
| IA$^3$ | 540K | 64.6 |
| w/pre-train | | |
| LoRA | 2.2M | 66.8 |
| TLoRA | 1.4M | 66.1 |
| Poly | 17M | 68.8 |
| TensorPoly-I(R=8,N=2) | 12.2M | 68.7 |
| TensorPoly-I(R=20,N=2) | 27.8M | 69.3 |
| TensorPoly-II(R=8,N=2) | 13.3M | 67.9 |
| TensorPoly-II(R=20,N=2) | 33.2M | 68.0 |

Table 7: We show the parameters of different approaches in the finetuning process. For the baselines, we report the results from the paper (Liu et al., 2022).

### A.3 Few-shot finetuning parameter size over different PEFT approaches

We investigate the training parameter size over PEFT approaches in the few-shot finetuning process. As presented in Tab. 7, TLoRA obtains competitive results while using only 60% training parameters than the original LoRA in this train setting. Compared with the latent experts approach `Poly`, `TensorPoly-I` with tensor rank $R = 8$ and order $N = 2$, get almost the same results with fewer finetuning parameters. `TensorPoly-I(R=20,N=2)` obtain the best results with 27.8 finetuning parameters. Notably, the finetuning parameters are not the adaption parameters we present in Tab. 3.

### A.4 Routing across different tasks

To investigate the routing function across different tasks, we plot the routing distributions for tasks such as Question Answering and Summarization. As depicted in Figure 6, the routing function distribution varies across different layers for each task. Notably, Closed-Book QA exhibits greater sparsity compared to multi-choice QA. We will conduct further analysis of the routing function in various tasks to gain deeper insights.

### A.5 Latent Experts Visualization

We visualize the latent experts in our models. As depicted in Figure 7, we draw the expert's weight distribution in layer [0,4,16,23]. The distribution of expert weights varies in different layers. Especially, The expert weights in layer 16 seem completely different from the experts in layers [0, 4, 23]. We also draw the similarity of expert weights across different layers in Figure 8.

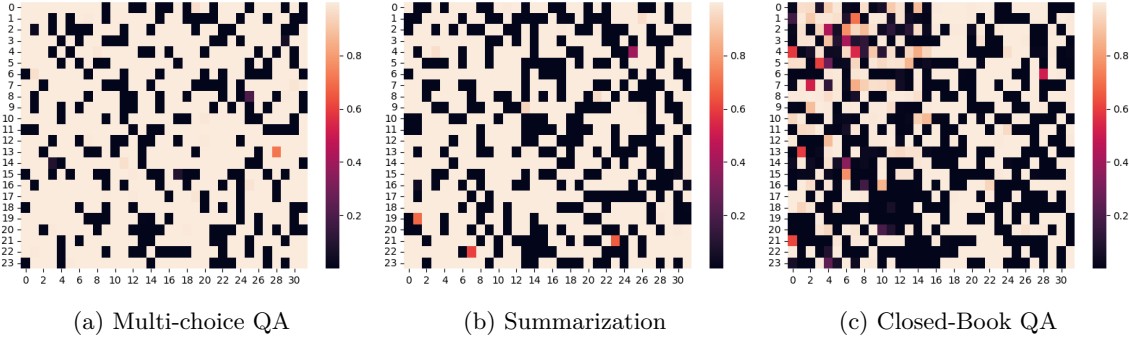

(a) Multi-choice QA          (b) Summarization          (c) Closed-Book QA

Figure 6: The X-axis is the 32 experts in each layer (For Q,K,V,O, each corresponds to 8 experts). Y-axis denotes the 24 transformer layers in the T0 model. We show routing distributions for different kinds of tasks.

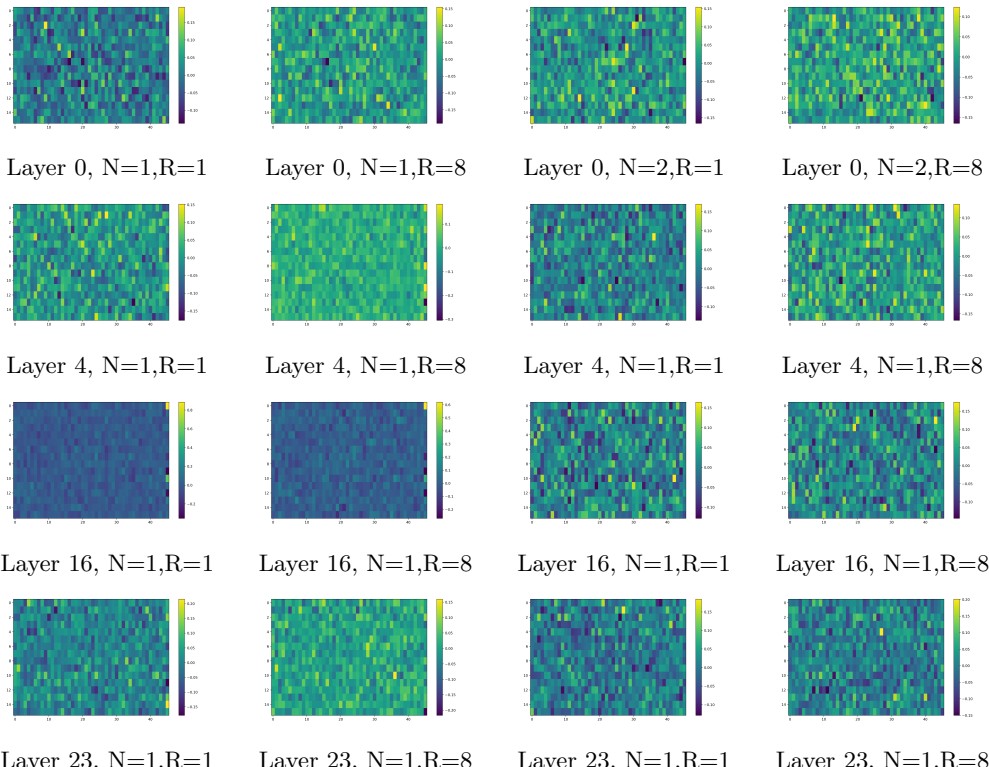

Figure 7: Weights distributions for latent experts across different layers. $N$ is the order and $R$ is the tensor rank. Each weight matrix here is $\mathcal{A}_{i,k} \in \mathbb{R}^{r \times \lceil \sqrt[N]{d} \rceil}$.

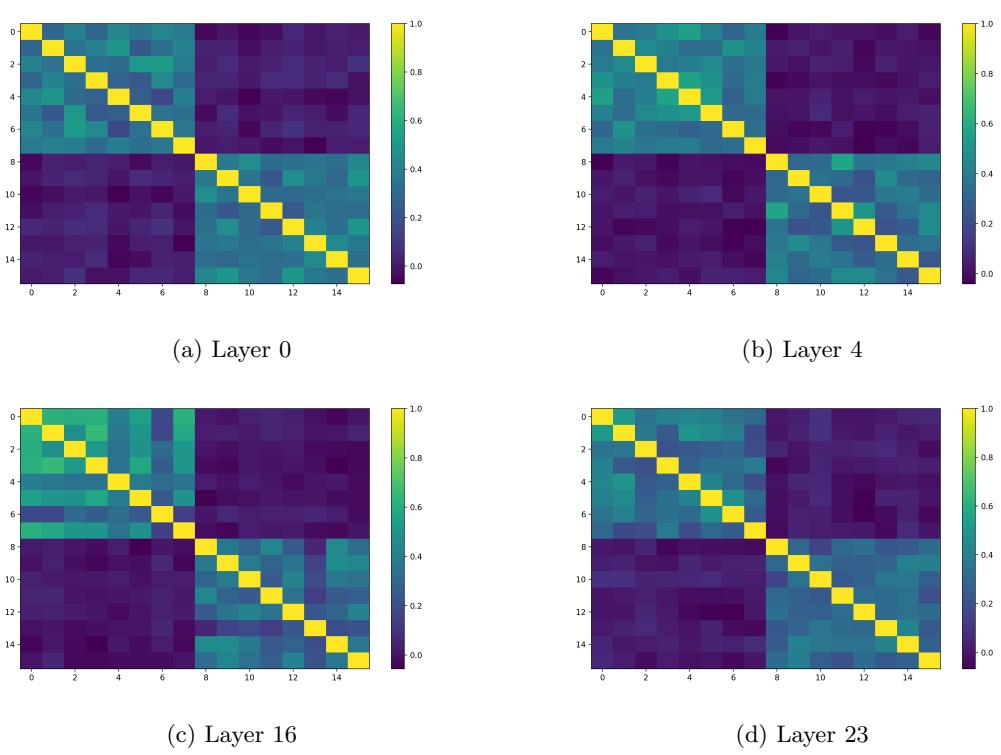

(a) Layer 0

(b) Layer 4

(c) Layer 16

(d) Layer 23

Figure 8: Expert similarity in each layer in `TensorPoly-II`. We compare the similarity of 16 experts in each layer. 0-7 refers to experts with order 1, 8-15 refers to the experts with order 2.

