# OpenReview forum: "Mixture of Latent Experts Using Tensor Products"
_TMLR — Accepted by TMLR_

### Review · Reviewer_ASZr · 2024-07-22

**Summary Of Contributions:**

This work proposes TLoRA, which is a parameter-efficient learning method that requires fewer tunable parameters than LoRA. This method also incorporates MoE, which improves the performance.

**Audience:**

Yes

**Claims And Evidence:**

Yes

**Requested Changes:**

1. It is not clear why the tensor-based method is better than LoRA. Is it because LoRA cannot capture some underlying structure but the proposed method can? Is there any intuitive explanation?

2. It is still not clear why TensorPoly-I performs better than TensorPoly-II. Any theoretical explanation?

3. I think Section 5.3.1 aims to compare the proposed method and LoRA. However, TensorPoly-I and TensorPoly-II include MoE in the architecture. It is unclear whether the performance improvement is due to MoE or the tensor construction.

**Strengths And Weaknesses:**

Strengths:
1. The proposed method is interesting and makes sense.
2. The paper is well-written.

Weaknesses:
1. The deeper understanding of the proposed method is lacking.
2. It is not clear whether the proposed method can be applied to LLM.

---

> ### Author Response · Authors · 2024-08-12
> **rebuttal**
>
> Thank you for your feedback, we address your concerns point by point.
>
> > Q1:  It is not clear why the tensor-based method is better than LoRA. Is it because LoRA cannot capture some underlying structure but the proposed method can? Is there any intuitive explanation?
>
> Thanks for your question. From our experimental results. We can show that routing-based methods can outperform the single adapter (poly can be better than the LoRA, TensorPoly can be better than the TLoRA). Our approach TensorPoly-I can achieve the best results in all routing-based methods. From Figure 5 in our paper, we can show that the routing-based methods can achieve the lowest validation loss compared to a single adapter.
>
> > Q2: It is still not clear why TensorPoly-I performs better than TensorPoly-II. Any theoretical explanation?
>
> Our best guess is that the object $\Delta W$ is essentially expressed as a cumulative product of parameterized terms $A_{1,k}, ..., B_{1,k}$ in TensorPoly-II, which may lead to unstable training dynamics and adds difficulty to model training. In contrast, TensorPoly-I conducts a weighted sum of tensors in both components (i.e., A, B) of $\Delta W$, which contributes to a stable training process.
>
> > Q3:  I think Section 5.3.1 aims to compare the proposed method and LoRA. However, TensorPoly-I and TensorPoly-II include MoE in the architecture. It is unclear whether the performance improvement is due to MoE or the tensor construction.
>
> Tensor construction builds a new kind of latent expert approach. If we only observe the tensor construction (TLoRA), the results are only on par with the LoRA with fewer adaption parameters. Please refer to the general response.
>
> > Q4: It is not clear whether the proposed method can be applied to LLM.
>
> Please see the answer to Q5 from reviewer 2.

---

### Review · Reviewer_Lh9h · 2024-07-29

**Summary Of Contributions:**

This paper explores improved ways to reduce cross-task interference in multi-task learning setup. To achieve this, the authors proposed to apply tensor-product based parameter efficient tuning to the mixture of expert optimization.

Technical contribution includes:
- Propose TLoRA that applies tensor product to LoRA, further reducing trainable parameters thus improving parameter efficiency;
- Propose TensorPoly that combines mixture of expert and TLoRA at different levels of granularity;

Under the T0 benchmark, TensorPoly achieves slightly better performance than previous models under a similar parameter budget.

**Audience:**

Yes

**Broader Impact Concerns:**

I didn't find serious issues.

**Claims And Evidence:**

No

**Requested Changes:**

- It seems that TensorPoly adds extra flops during the training. Please add the amount of trainable parameters and extra training flops as two columns into Table 1 for better comparison.
- I don’t understand why the authors name the method “mixture of latent experts”. I didn’t see any relationship between tensor products and the term “latent”. The used MOE method seems pretty standard.
-  Apart from token-level MOE, there are also many task-level MOE methods that are not discussed and compared in this paper, such as [1] and [2].
- I’m curious whether we need to select a subset of experts in the multi-task learning setup. What if apply soft selection, i.e. directly constructing a softmax distribution over the experts?
- Scaling becomes an important aspect for model finetuning [3]. How does the increase of model size, task number, and LoRA rank affect TensorPoly? Should we expect that TensorPoly scales reasonably? A concern is that tensor products may hurt training stability.
- In Figure 3 left, what’s the number of rank corresponding to the similar amount of parameters to the standard LoRA. It seems that further scaling up the number of ranks beyond 20 hurts the performance; does this suggest TensorPoly has limited scaling property?
- In Table 1, TensorPoly often performs worse on co-reference and some NLI tasks. Any analysis why?
- In page 9, the authors claim “TensorPoly-I stands out by not only surpassing recent state-of-the-art achievements”. In fact, TensorPoly only outperforms MHR by 0.1 point on average (and underperforms it on several tasks), which is very likely not significant at all. Such a claim may be inappropriate.

[1] Kudugunta et al., Beyond distillation: Task-level mixture-of-experts for efficient inference

[2] Baziotis et al., Multilingual Machine Translation with Hyper-Adapters

[3] Zhang et al., When scaling meets llm finetuning: The effect of data, model and finetuning method

**Strengths And Weaknesses:**

Strengths:

TensorPoly improves parameter efficiency and achieves better multi-task learning performance than its competitors;


Weaknesses:
- TensorPoly relies on tensor-product which introduces extra training flops and may increase the risk of training instability
- The quality improvement over MHR is very marginal, particularly considering their similar parameter efficiency; the authors may need to figure out a clear setup where TensorPoly truly stands out.
- More baselines and comparisons should be added
- Scaling experiments is missing

---

> ### Author Response · Authors · 2024-08-12
> **rebuttal**
>
> We thank the reviewer for providing a thorough and constructive review. We will address the questions one by one.
>
> > Q1: TensorPoly relies on tensor-product which introduces extra training flops and may increase the risk of training instability.
>
> Thanks for your suggestions. Yes, the tensor-product introduces extra training flops just like any other tensor network approach.  We add some flops analysis in the Appendix. The tensorized methods will result in a little longer training time compared to other baselines, which will lead to a minimal increase in the risk of training instability to the authors’ best knowledge.
>
> > Q2: I don’t understand why the authors name the method “mixture of latent experts”. I didn’t see any relationship between tensor products and the term “latent”. The used MOE method seems pretty standard.
>
> Sorry for the confusion. The latent experts are not based on tensor products. We present a comparison picture in Figure 2 in the revised version.  For |T| tasks, Compared to the private version, where number of experts |K|=|T|, and the shared version |K|=1, the latent expert approach is much more like a trade-off, where the number of experts |K|<|T|. In the latent experts' approach, we posit that there exists a (possibly small) fixed inventory of experts, Each expert is an independent facet of knowledge that is reused across a subset of tasks.
>
> > Q3: Apart from token-level MOE, there are also many task-level MOE methods that are not discussed and compared in this paper, such as [1] and [2].
>
> Thanks for your suggestions, we have added these two papers to the related works. Actually, in the paper of Poly, they have discussed these two task-level baselines(Task-MOE and hyper-former). In the few-shot scenarios, the Poly outperforms these two baselines.
>
> > Q4: I’m curious whether we need to select a subset of experts in the multi-task learning setup. What if apply soft selection, i.e. directly constructing a softmax distribution over the experts?
>
> Good question, when we apply a soft selection directly. For each task,  we need to get a LoRA adapter, in our case, 313 tasks will result in 313 LoRA adapter which is not highly parameter efficient. In addition, this baseline (adapterSoup) is studied in the paper MHR, the results show that adapter soup underperforms the latent expert approach such as Poly/MHR.
>
> The MHR paper explores whether adding more LoRA adapters can contribute to the improvement. In Figure 3 right of MHR, when adding more experts, the results stay the same, indicating that there is a tradeoff between the number of experts and the performance.
>
> > Q5: Scaling becomes an important aspect of model finetuning [3]. How does the increase in model size, task number, and LoRA rank affect TensorPoly? Should we expect that TensorPoly scales reasonably? A concern is that tensor products may hurt training stability.
>
> We have tested TensorPoly-I and LoRA on the 11B T0 model. We got LoRA with 72.3 and TensorPoly with 73.8 (See table below). The experimental results show that Tensorpoly can outperform the LoRA no matter in 3B and 11B. We will test other backbone models in  further investigations.
>
> | Model             | Average ACC |
> |-------------------|-------------|
> | LoRA (3B)         | 66.8        |
> | TensorPoly-I (3B) | 69.3        |
> | LoRA (11B)        | 72.3        |
> | TensorPoly-I (11B)| 73.8        |
>
> > Q6:  In Figure 3 left, what’s the number of ranks corresponding to a similar amount of parameters to the standard LoRA? It seems that further scaling up the number of ranks beyond 20 hurts the performance; does this suggest TensorPoly has limited scaling properties?
>
> When we set the rank=2, Tensorpoly has a comparable number of training parameters as ]LoRA, with a performance value of 67.7 (TensorPoly) compared to 66.8 (LoRA). When we set a larger rank, it does not necessarily lead to a better performance As shown in the table, there may be an overfitting issue when rank>=30.
>
> | Rank | Training parameters       | Average ACC |
> |------|---------------------------|-------------|
> | 1    | 1.4M                       | 66.1        |
> | 2    | 2.8M (LoRA, 2.2M)          | 67.7        |
> | 3    | 4.2M                       | 67.7        |
> | 8    | 12.2M                      | 68.7        |
> | 20   | 27.8M                      | 69.3        |
> | 30   | 41.8M                      | 69.1        |

---

> ### Author Response · Authors · 2024-08-12
> **rebuttal**
>
> > Q7: In Table 1, TensorPoly often performs worse on co-reference and some NLI tasks. Any analysis of why?
>
> Thanks for your question, it is difficult to show an explicit reason. We study the adapter similarity in co-reference tasks Winogrande and WSC after finetuning. For 20 experts, we compute the cosine similarity of adapter pairs (20 * 19 / 2), and the results show that Poly has more diversity compared to TensorPoly.
>
> | model         | Winogrande | WSC |
> |---------------|--------|---------|
> | Poly         | -0.94    | -0.95   |
> | TensorPoly-I  | 41.90 | 41.91   |
>
> > Q8: On page 9, the authors claim “TensorPoly-I stands out by not only surpassing recent state-of-the-art achievements”. TensorPoly only outperforms MHR by 0.1 point on average (and underperforms it on several tasks), which is very likely not significant at all. Such a claim may be inappropriate.
>
> Thanks for your suggestion. We have updated the text.
>
> > Q9: It seems that TensorPoly adds extra flops during the training. Please add the amount of trainable parameters and extra training flops as two columns into Table 1 for better comparison.
>
> Thanks for your suggestion. We did not know the training time of all baselines in Table 1. As a result, we create Table 6 for a better comparison in our experimental setting.

---

### Review · Reviewer_gFt3 · 2024-07-30

**Summary Of Contributions:**

This paper proposes a tensor-based method for modeling "latent expert" to achieve a balance between computational efficiency and knowledge transfer for multi-task learning. The experimental results show the effectiveness and efficiency.

**Audience:**

Yes

**Claims And Evidence:**

Yes

**Requested Changes:**

1. This paper shows the proposed method works. But it fails in telling us more about why it works. Please give more discussion on the mechanism.
2. It seems TensorPoly-II is more expressive than TensorPoly-I. However, it does not outperform TensorPoly-I in the experimental settings. It is better that the authors tell us the reason.

**Strengths And Weaknesses:**

Strength:
- A practical method for multi-task learning which is both efficient and effective.
- Novel applications of tensor model in multi-task learning.

Weakness

- The authors did not clearly explain why tensorized latent experts can alleviate negative transfers in multi-task learning.
- Kronecker-product-based weight factorization is parametric efficient. However, it may severely decrease the approximation capability, which corresponds to expressiveness of the model. The balance between efficiency and expressiveness is in general hard to achieve. I think more theoretical research needs to be done for this issue.

---

> ### Author Response · Authors · 2024-08-12
> **rebuttal**
>
> Thank you for your feedback. Below we address your concerns point by point.
>
> > Q1: This paper shows the proposed method works. But it fails to tell us more about why it works. Please give more discussion on the mechanism.
>
> As is explained above in the general response, TensorPoly is able to alleviate negative transfers as a latent expert model. Moreover, it outperforms existing latent expert models in that it brings about high-diversity adapter weights across different tasks, which further promotes positive transfers.
>
> > Q2: It seems TensorPoly-II is more expressive than TensorPoly-I. However, it does not outperform TensorPoly-I in the experimental settings. It is better that the authors tell us the reason.
>
> Thanks for your question, We will do more analysis on this. As shown in Figure 5, the TensorPoly-II has more training parameters and complex routing functions, while the relation between latent experts and real tasks is relatively opaque and intractable. Possibly due to this, the validation loss of TensorPoly-II is higher than the TensorPoly-I, indicating that it does not show a higher expressivity across all tasks. As discussed in the paper (5.4.2), we plan to explore the circumstances under which the tensor-product interaction unique in TensorPoly-II can benefit.

---

### Decision · Action_Editor_Yf1Y · 2024-09-16

**Recommendation:** Accept with minor revision

**Comment:**

After the rebuttal, the overall rating of this paper is positive. The reviewers acknowledge that the proposed method is ''efficient and effective'' have ''novel applications in multi-task learning''. The rebuttal is sufficient and has addressed most of the concerns. However, the remaining issue is about the clarification of computation cost and parameter size. Thus, this paper should be carefully revised before acceptance.

**Audience:**

The researchers in muti-task learning and deep learning will be interested in the findings of this paper.

**Claims And Evidence:**

The claims of this work are supported by the comparison experiments and ablation study.

---

> ### Author Response · Authors · 2024-10-15
> **Official Comment by Authors**
>
> Dear Action Editor,
>
> Thanks for your positive feedback and suggestions. we have already made the following revisions to the latest version of our manuscript:
> (1) We add the Flop analysis section in the appendix (Page 18) and compare the training time in the pretraining process.
> (2) We present the parameter size in the finetuning process for all PEFT methods used in our setting in the appendix (Page 19).
>
> We have carefully revised the manuscript according to the reviewer's suggestions and believe these changes have improved the clarity and robustness of our paper. We hope the revised version addresses the concerns raised and meets the high standards of TMLR.
>
> Thank you once again for your guidance and consideration. We look forward to the final decision.